# Golgi reassembly and stacking protein 65 downregulation is required for the anti-cancer effect of dihydromyricetin on human ovarian cancer cells

**Fengjie Wang** [1,2], **Xianbing Chen** [2], **Depei Yuan** [2], **Yongfen Yi** [1], **Yi Luo** [1,3]*

**1** Department of Pathology, College of Basic Medicine, Chongqing Medical University, Chongqing, China, **2** Minda Hospital of Hubei Minzu University, Enshi, Hubei, China, **3** Department of Gynecology and Obstetrics, The First Affiliated Hospital Of Chongqing Medical University, Chongqing, China

* smileyi@163.com

**Data Availability Statement:** All relevant data are within the manuscript and its Supporting Information files.

## Abstract

Golgi reassembly and stacking protein 65 (GRASP65), which has been involved in cancer progression, is associated with tumor growth and cell apoptosis. Dihydromyricetin (DHM) has demonstrated antitumor activity in different types of human cancers. However, the pharmacological effects of DHM on ovarian cancer (OC) and the molecular mechanisms that underlie these effects are largely unknown. The present study showed that DHM reduced cell migration and invasion in a concentration- and time-dependent manner and induced cell apoptosis primarily through upregulation of Cleaved-caspase-3 and the Bax/Bcl-2 ratio in OCs. To further clarify the cancer therapeutic target, we assessed the effect of DHM on the expression of GRASP65, which is overexpressed in human ovarian cancer tissues. DHM activated caspase-3 and decreased GRASP65 expression to promote cell apoptosis, implying that downregulation of GRASP65 was related to DHM-induced cell apoptosis. Additionally, the knockdown of *GRASP65* by siRNA resulted in increased apoptosis after DHM treatment, while western blot and flow cytometry analysis demonstrated that overexpression of *GRASP65* attenuated DHM-mediated apoptosis. In addition, the JNK/ERK pathway may be involved in DHM-mediated caspase-3 activation and GRASP65 downregulation. Taken together, these findings provide novel evidence of the anti-cancer properties of DHM in OCs, indicating that DHM is a potential therapeutic agent for ovarian cancer through the inhibition of GRASP65 expression and the regulation of JNK/ERK pathway.

## Introduction

Electron microscopy has been used to demonstrate Golgi fragmentation (GF) in tumor cells [1], and we have only just begun to understand the significance of GF in tumor biology. GF serves as a catalyst for the cell signaling pathways that drive cancer progression and metastasis. However, the causal relationship between GF and cancer pathogenesis remains largely unexplored. For example, swainsonine, an inhibitor of Golgi alpha-mannosidase II, has been

**Funding:** This work was supported by the Health and Family Planning Commission of Hubei Province (No. WJ2019M104).

**Competing interests:** The authors have declared that no competing interests exist.

shown to have antitumor activity in gastric carcinoma [2]. Another anti-Golgi agent, Brefeldin A, showed antiproliferative effects *in vitro* and inhibition of tumor growth *in vivo* [3]. Golgi reassembly and stacking proteins (GRASPs) are Golgi membrane proteins involved in cell migration, division, and apoptosis. Specifically, GRASP65, a target of polo-like kinases (PLK1) and Cdc2 during mitosis [4,5], mediates Golgi morphological changes to fulfill physiological functions [6–8]. In addition, the upregulation of Golgi proteins has been observed in many types of tumors, including ovarian cancer (OC). Golgi phosphoprotein3L (GOLPH3L) was overexpressed in epithelial ovarian cancer (EOC) tissues and cell lines [9] and associated with poor prognosis of patients with EOC [10]. GOLPH3 may promote EMT progression through the activation of Wnt/β-catenin pathway and act as a novel and independent prognostic factor of EOC [11]. Furthermore, silencing *GM130* decreased angiogenesis and cell invasion *in vitro* and in a lung cancer mouse model, suggesting that it may be a potential therapeutic target for lung cancer [12]. Restoration of compact Golgi morphology in advanced prostate cancer may increase the susceptibility to Galectin-1-induced apoptosis [13], strengthening the notion of the "oncological Golgi" and its role in cancer progression and metastasis [1]. Therefore, targeting the Golgi proteins may be a potential therapeutic intervention for multiple cancers [14].

OC is one of the most common gynecological malignancies with high rates of metastasis and disease relapse worldwide. The invasion and progression of OC cells are presumed to be a multistep process involving multiple genetic changes. Consequently, numerous studies have focused on the identification of specific molecular markers that may serve as reliable prognostic biomarkers for ovarian cancer. Additionally, the current standard of care treatment for patients with ovarian cancer is surgery coupled with platinum and/or Taxane-based chemotherapy. While most patients are initially responsive to chemotherapy, the 5-year survival rate of OC patients is approximately 15–30% [15]. Therefore, there is an urgent need to improve the techniques employed for early disease detection, and to identify effective therapies to improve clinical outcomes for OC patients.

Recently, researchers have turned their attention to natural active compounds extracted from medicinal plants for the treatment of cancer patients [16]. Most natural compounds have shown cytotoxicity only in cancerous cells and are therefore potential therapeutic agents for future clinical development [17]. In addition, several studies have demonstrated that these components can substantially inhibit tumor formation and induce apoptosis [18,19]. Dihydromyricetin (DHM), a 2,3-dihydroflavonol compound, is the main bioactive component extracted from *Ampelopsis grossedentata* [20] and has attracted considerable attention in cancer research for its antitumor effects [21–23]. DHM has been shown to be an effective anticancer agent in various cancers and is also considered to have great antitumor potential for the treatment of OC [24]. However, the mechanism underlying the antitumor effect of DHM needs to be investigated.

In response to stress, the transcription of Golgi-associated genes can be upregulated to restore homeostasis or induce apoptosis, which gave rise to the term *Golgi stress response* (GSR) [25,26]. The role of GSR and cell apoptosis in chemotherapy can be quite complex [27] and their connection has made them an intriguing target that may improve anti-cancer treatment. Furthermore, morphological studies have shown that the Golgi complex is fragmented during apoptosis [28], and GF in apoptotic cells may be attributed to GRASP65 cleavage [29]. GRASP65 is phosphorylated by Cdc2 and PLK-1 during cell mitosis, which leads to GRASP65 deoligomerization and then Golgi unstacking [5,30]. Additionally, as a potential small molecular inhibitor of PLK-1, DHM may prevent cancer progression by inhibiting PLK-1 enzymes [31]. Therefore, we hypothesized that DHM possesses anti-tumor activity by regulating GRASP65 function. We also investigated the mechanisms and effects of DHM on OCs in order to provide preliminary evidence for future clinical applications.

## Materials and methods

### Reagents

Dihydromyricetin (CAS No. 27200-12-0, Bellancom) was ordered from Beijing Universal Materials Co., Ltd. (Beijing, China), with purity >98%, as detected by high performance liquid chromatography. DHM was dissolved in 100% dimethyl sulfoxide (DMSO) to prepare a 50 mM stock solution and was stored at −20˚C. DHM solutions used in cell cultures were freshly prepared daily and the final concentration of DMSO did not exceed 0.1% throughout the study.

Apoptotic cells were quantified using an Annexin V-FITC/PI cell apoptosis detection kit from Becton Dickinson and Company (Franklin Lakes, NJ, USA) and monitored using flow cytometry (FACSCalibur, BD, Franklin Lakes, NJ, USA).

JNK inhibitor SP600125, ERK inhibitor U0126, and Caspase-3 inhibitor Ac-DEVD-CHO were purchased from Beyotime (Shanghai, China) and dissolved in DMSO at concentrations of 20, 10 and 30 μM, respectively.

Antibodies for p-JNK/JNK, p-ERK/ERK, and GRASP65 were bought from Abcam (Cambridge, MA, USA), antibody for Actin from Beyotime (Shanghai, China), and antibodies for p-p38/p38MAPK, cleaved-caspase-3, Bcl-2, and Bax from Cell Signaling Technology (Danvers, MA, USA).

### Cell culture

The human ovarian cancer SKOV3 cell line was purchased from Boster Biological Technology Co., Ltd. (Wuhan, China). The A2780 cell line was obtained from the Molecular Medicine and Cancer Research Center of Chongqing (Chongqing, China) and cultured in DMEM (Hyclone, Logan, Utah, USA), and supplemented with 10% FBS (Gibco, Invitrogen Life Technologies, Carlsbad, USA), 100 unit/mL penicillin, and 100 mg/mL streptomycin (Beyotime, Shanghai, China) in a humidified chamber containing 5% $CO_2$ at 37˚C.

### In vitro cell viability assay (CCK8)

SKOV3 and A2780 cells were seeded in 96-well microtiter plates (Corning, NY, USA) with $1 \times 10^4$ cells per well and pretreated with various concentrations of DHM for 24 h and 48 h to select the most effective concentration and time point for the assessment of cell viability using the Cell Counting Kit-8 (Beyotime, Shanghai, China) following the manufacturer's recommendations. Six reduplicate wells were used for each treatment and the experiments were performed three times. At each time point, the absorbance ($A$) was measured at 450 nm using a microplate reader (BioTek synergy HT, VT, USA). The concentration required to inhibit cell growth by 50% ($IC_{50}$, half maximal inhibitory concentration) was calculated using GraphPad Prism (San Diego, CA, USA).

The percentage of viable cells was calculated as follows:

$$\text{Cell viability}(\%) = (A_{\text{treated}}/A_{\text{control}}) \times 100\%.$$

### Wound healing assay

Wound healing assay was performed and the closure of the scratched area was calculated as previously described [32]. Cells were seeded at a high density ($1 \times 10^6$ cells/mL) in each well of a 6-well culture plate and allowed to adhere. Confluent cells were scratched with a 200 μL pipette tip and treated with the various concentrations of DHM diluted by serum-free DMEM

medium and were cultured for the indicated time points. Cells were then photographed with a digital camera (IX70, Olympus) and the wound width was measured using an image analysis software (ImagePro Premier). Three fields were randomly selected from each wound.

## Migration/Invasion assay

Cell migration assay was performed and determined using Corning Transwell insert chambers (Cat No. 3422, Corning, NY, USA). Thaw the Matrigel® Matrix (Becton Dickinson, Oxford, UK; Cat No. 354234) overnight at 4°C and mix Matrigel with serum-free cold DMEM. 0.1ml of the diluted Matrigel was pre-coated directly onto each 24-well Transwell insert at 37°C for at least 1h for invasion assay [33].

## Apoptosis assay

Apoptotic cells were assessed using an Annexin V-FITC/PI kit (BD Pharmingen, Franklin Lakes, NJ, USA) according to the manufacturer's instructions. Apoptotic cells were detected by flow cytometry (FACSCalibur, Becton Dickinson, San Jose, CA, USA) and analyzed using FlowJo cell analysis software (FlowJo, Ashland, OR, USA).

## Caspase-3/9 activity assay

Caspase-3/9 activity was measured using a commercial kit (Beyotime, Shanghai, China) according to the manufacturer's instructions. After treatment with different concentrations of DHM, cells were lysed and then incubated with the caspase reagent and its substrates, Ac-DEVD-pNA (caspase-3) and Ac-LEHD-pNA (caspase-9), for 1–2 h at 37°C. Absorbance ($A$) at 405 nm was measured using a microplate reader. The caspase activity (Unit) = $A_{\text{treated}}$ / $A_{\text{control}} \times 100\%$.

## Cellular immunofluorescence (IF)

Cells were seeded on slides in a 24-well plate and allowed to attach for 24 h. The cells were subsequently treated with different concentrations of DHM. For IF analysis, the samples were fixed with 4% paraformaldehyde and permeabilized with 0.5% Triton X-100 for 10 min. After being blocked with 10% goat serum albumin for 60 min at 37°C, the slides were incubated with primary antibodies against Caspase-3 (1:500, Rabbit mAb), Actin (1:400, mouse mAb) and GRASP65 (1:300, Rabbit mAb) overnight at 4 °C. The following day, the slides were washed three times with PBS and then incubated with FITC-labeled Goat anti-mouse or Cy3-labeled Goat anti-rabbit secondary antibodies for 1 h at 37°C before being labelled with DAPI for 1 min at room temperature. Finally, three fields per slide were randomly selected for observation under a fluorescence microscope (Olympus Inc., Tokyo, Japan). Staining intensities were measured by observers blinded to the experimental groups using Image-Pro Plus 6.0 (Media Cybernetics, Rockville, MD, USA).

## Gene knockdown using *siRNA* and transient transfection

Two GRASP65 siRNAs (siRNA1: `GGUUGGUUCGGACCAGAUUTT`; siRNA2: `GGAACCAUCUUC ACCUGCUTT`) and one overexpression plasmid (NCBI Reference Sequences of GRASP65/ GORASP1(human): NM_031899.3), as well as the corresponding negative control plasmids, were all designed and synthesized by Shanghai GenePharma Co., Ltd. Cells were transiently transfected with siRNAs or plasmid using the Lipofectamine® 2000 Reagent kit (Invitrogen; Thermo Fisher Scientific, Inc., MA, USA) according to the manufacturer's protocol.

Following a 6 h transfection, the cell culture solution was changed to a normal medium. After transfection for 24–48 h, cells were processed for further analysis and subsequent experiments, and non-transfected cells served as blank controls. Transfection efficiency was verified using western blot analysis.

## Preparation of total cell extracts and western blot analysis

The cells were lysed with lysis buffer containing protease inhibitors. The soluble cell lysates were collected after centrifugation at 14000 g for 15 min. Equal amounts of protein (20–30μg) were subjected to 10 or 12% sodium dodecylsulfate polyacrylamide gel electrophoresis (SDS-PAGE) and transferred onto polyvinylidene fluoride (PVDF) membranes (Millipore, Belford, MA). After blocking with 5% skim milk in PBS containing 0.1% Tween20 (TBST), membranes were incubated with primary antibodies at 4 ˚C overnight. After washing with TBST three times for 10 min, the membranes were incubated at room temperature for 2 h with secondary antibodies. An enhanced chemiluminescent substrate (ECL, Thermo fisher, MA, USA) was added to the membranes, which were photographed using a protein analysis system (Tanon 5200, Shanghai, China).

## Statistical analysis

All data are presented as the mean ± standard deviation (SD). The statistical significance of differences between groups was analyzed by one-way Analysis of Variance (ANOVA) followed by Dunnett's or Tukey's post hoc tests using SPSS 17.0 software (SPPS Software, Inc., Chicago, IL, USA). A value of $P < 0.05$ was considered statistically significant.

## Results

### DHM reduced cell migration and invasion in SKOV3 and A2780 cells

The inhibitory effects of DHM on A2780 and SKOV3 ovarian cancer cells were assessed using CCK8, wound healing, and Transwell assays. A previous study showed that DHM had no significant cytotoxicity in human ovarian surface epithelial cells [24]. Firstly, the cell viability was detected by CCK-8 assay after treatment with DHM. DHM treatment significantly decreased the cell viability of SKOV3 and A2780 cells in a time- and dose-dependent manner. The $IC_{50}$ values of DHM for SKOV3 and A2780 cells were 213.4 and 157.2 μM, respectively, after DHM treatment for 24 h. The $IC_{50}$ values were 132.3 and 98.2 μM, respectively, after DHM treatment for 48 h. We selected 120 and 80 μM of DHM for 48 h in SKOV3 and A2780 cells based on the determined $IC_{50}$ values for subsequent studies, respectively.

Next, to examine whether DHM inhibited the migration of ovarian cancer cells, wound healing assays were performed using non-cytotoxic concentrations of DHM. The closure of the wound in the DHM-treated group was calculated and normalized to that of the control. As shown in Fig 1A and 1B, 40 μM DHM suppressed approximately 50% of wound closure in A2780 cells, and significantly reduced the closure of the wound after DHM treatment at both 80 μM and 120 μM in SKOV3 cells.

Additionally, Transwell assays were carried out to confirm the inhibitory effect of DHM on cell migration. The number of migratory DHM-treated cells was calculated and then normalized to that of the control cells. As shown in Fig 1A and 1B, DHM dose-dependently inhibited the migration of both ovarian cancer cell lines. Transwell assays were also performed to explore the effect of DHM on the invasion of cancer cells. Our results clearly showed that DHM significantly attenuated the invasion of SKOV3 and A2780 cells.

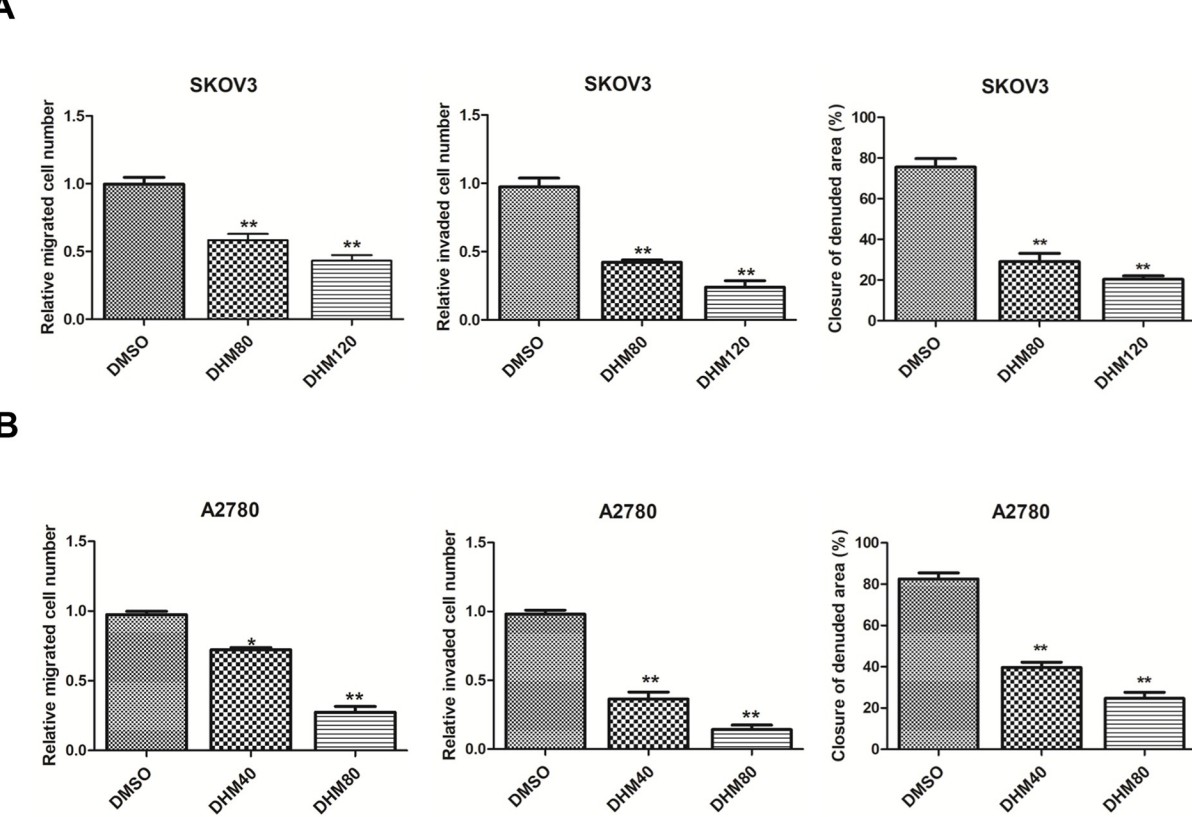

**Fig 1. DHM reduces cell migration and invasion. A/B**. Wound healing and Transwell assay were conducted in. SKOV3 cells (**A**) treated with DMSO (control), 80, and 120 μM of DHM, and A2780 cells (**B**) with DMSO (control), 40, and 80 μM for 48 h. Each experiment was repeated at least three times. * p < 0.05, **p < 0.01 vs the DMSO (control) group.

These results suggest that DHM could decrease cell viability and reduce the migration and invasion of different ovarian cancer cells at non-cytotoxic doses, implying that DHM may be a potent therapeutic agent for ovarian cancer.

## DHM induced cell apoptosis in A2780 and SKOV3 cells

The apoptotic process is executed by a member of the highly conserved caspases, and modulation of the mechanisms of caspase activation and suppression is a critical molecular target in chemoprevention, since these processes lead to apoptosis [34]. To identify the mechanisms, cell nuclei were evaluated after DHM treatment by DAPI and Caspase-3 (Red) and Actin (Green) double staining was observed using fluorescence microscopy. Activation of Caspase-3/9 was also detected.

A2780 and SKOV3 cells were treated with 80 and 120 μM DHM, respectively, for 48 h to test the effects of DHM on cell apoptosis. DHM treatment resulted in an increase in nuclear disassembly, chromatin condensation, and the expression of Caspase-3 using IF double staining (Fig 2A), and the activation of caspases (Caspase-3 and -9) (Fig 2B), indicating apoptotic cell death in SKOV3 and A2780 cells.

Upon further investigation, we performed western blot analysis to evaluate the expression of apoptosis-related proteins following DHM treatment. As shown in Fig 2C, exposure to 80 μM DHM for 48 h resulted in an increase in Bax and cleaved-caspase-3 protein levels, and a

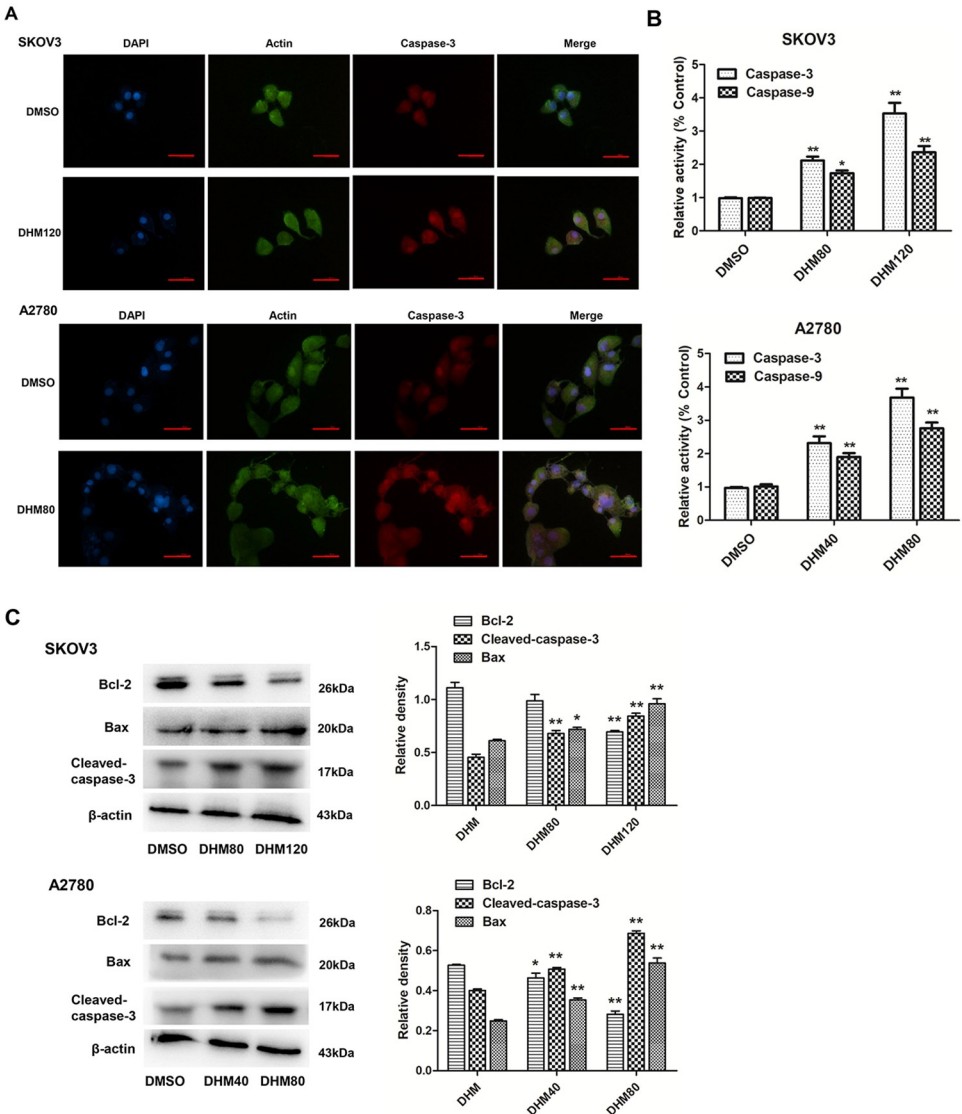

**Fig 2. DHM induces apoptosis in SKOV3 and A2780 cells. A**. SKOV3 cells were treated with 120 μM DHM and A2780 with 80 μM for 48 h. Caspase-3 (Red) and Actin (Green) double staining were characterized by IF staining and observed under fluorescent microscopy. **B**. Caspase-3/9 activity was determined after exposure to DHM for 48 h. **C**. Expression of Bcl-2, cleaved-caspase-3, and Bax was determined by western blot analysis. Scale bar equals 50 μm. Each experiment was repeated at least three times. * p < 0.05, **p < 0.01 vs the control group.

decrease in the expression of Bcl-2 in A2780 cells. Similar DHM-induced apoptotic effects were also observed in SKOV3 cells (Fig 2C).

These results suggest that DHM can induce apoptosis in ovarian cancer cells.

## DHM downregulated GRASP65 expression in SKOV3 and A2780 cells

We examined the expression of GRASP65 to determine its role in the inhibitory effect of DHM in ovarian cancer cells. SKOV3 and A2780 cells were treated with DHM for 48 h, and the expression of GRASP65 was detected by IF staining and western blot. In comparison to the control group, DHM induced cell apoptosis as shown in Fig 2, followed by the downregulation of GRASP65 expression in a concentration-dependent manner in Fig 3C and 3D. This finding

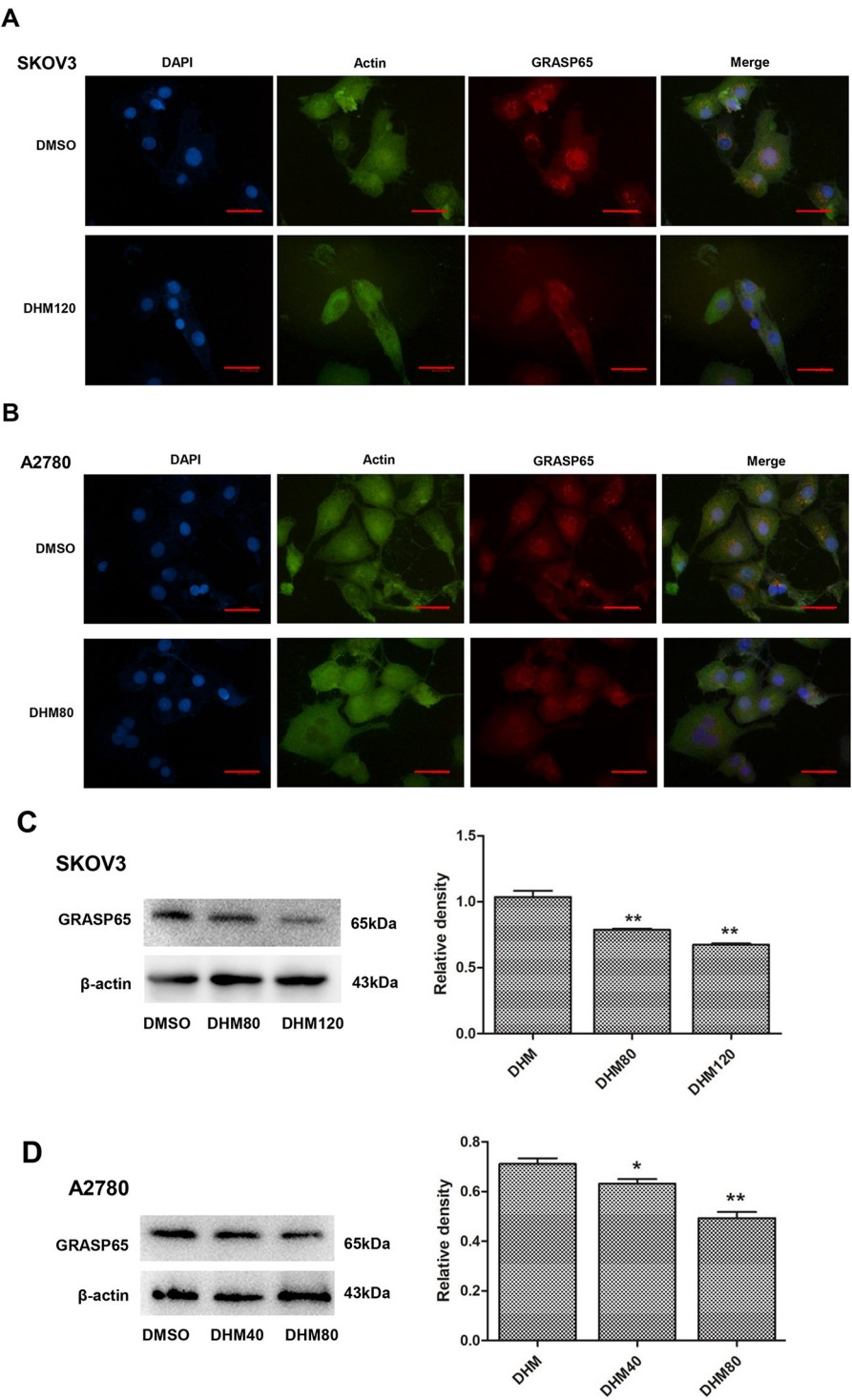

**Fig 3. DHM downregulates GRASP65 expression in SKOV3 and A2780 cells. A**. IF analysis of GRASP65 expression in DHM-treated SKOV3 cells. **B**. IF analysis of GRASP65 expression in DHM-treated A2780 cells. **C**. Western blot analysis of GRASP65 expression in DHM-treated SKOV3 cells. **D**. Western blot analysis of GRASP65 expression in DHM-treated A2780 cells. Scale bar equals 50 μm. Each experiment was repeated at least three times. $^*$ p < 0.05, $^{**}$p < 0.01 vs the control group.

was further supported by IF analysis (Fig 3A and 3B), indicating that DHM might downregu-late the expression of GRASP65 during DHM-induced apoptosis.

## DHM-induced caspase-3 activation was crucial for suppression of GRASP65 expression in SKOV3 and A2780 cells

Morphological studies have shown that the Golgi complex is fragmented and GRASP65 is cleaved by caspase-3 during apoptosis [29]. Additionally, the expression of a caspase-resistant form of GRASP65 partially preserved cisternal stacking and inhibited the breakdown of the Golgi ribbon in apoptotic cells [29]. To further explore the significance of caspase-3 activation and the relationship between caspase-3 activation and GRASP65 suppression in DHM-induced cell apoptosis, OCs were pre-treated with a specific caspase-3 inhibitor, Ac-DEVD-CHO, for 30 min to suppress the activity of caspase-3 to evaluate the contribution of caspase-3 in the effects of DHM. As shown in Fig 4A, Ac-DEVD-CHO dramatically attenuated DHM-induced increases in cleaved caspase-3, subsequently resulting in an increase in GRASP65 in SKOV3 and A2780 cells. This finding was further supported by flow cytometry analysis (Fig 4B).

The present results revealed that activated caspase-3 was crucial for suppression of GRASP65 in DHM-induced cell apoptosis. Therefore, we speculated that suppression of GRASP65 might be related to caspase-3 cleavage during DHM-mediated cell apoptosis.

## Effects of GRASP65 on DHM-induced cell apoptosis in A2780 cells

Previous studies have reported that GRASP65 is involved in cancer cell migration [35], polar-ity, and apoptosis [36]. To further determine whether DHM triggered cell apoptosis by decreasing GRASP65 expression, we silenced or overexpressed *GRASP65* in A2780 cells to determine the role of GRASP65 in DHM-induced cell apoptosis. The efficacy of transfection was confirmed by western blot analysis as shown in Fig 5A. Therefore, we selected the most effective siRNA2 to silence and OE plasmid to overexpress *GRASP65* in the following experiments.

As shown in Fig 5B, cells transfected with siGRASP65 showed a lower expression of GRASP65 and a higher level of cleaved caspase-3 than those in the control group, implying that GRASP65 depletion might lead to apoptosis in A2780 cells. Meanwhile, the expression of GRASP65 in cells transfected with siGRASP65 was lower than that in the control group when treated with 80 μM of DHM and the expression of cleaved caspase-3 was higher. The results of Annexin V-FITC/PI dual staining showed that there was more apoptosis in cells transfected with *GRASP65* siRNA in comparison to controls when treated with 80 μM of DHM (Fig 5B). The finding suggests that combination of DHM and GRASP65 depletion could further pro-mote apoptosis.

On the contrary, the level of cleaved caspase-3 decreased in the DHM-treated OE-GRASP65 group in comparison to the DHM-treated group (Fig 5C). Interestingly, the percentage of apo-ptotic cells was reduced to 8.59% in GRASP65-overexpressing cells after DHM treatment for 48 h, which was 5.94% lower than that observed in DHM group (Fig 5C). The above result revealed that overexpression of *GRASP65* attenuated DHM-mediated apoptosis.

## Effects of GRASP65 on DHM-mediated cell viability and migration in A2780 cells

Additionally, we also tested the effects of GRASP65 on cell viability and migration by CCK8 assay and wound healing assay, respectively, using *GRASP65* siRNA and OE-*GRASP65*

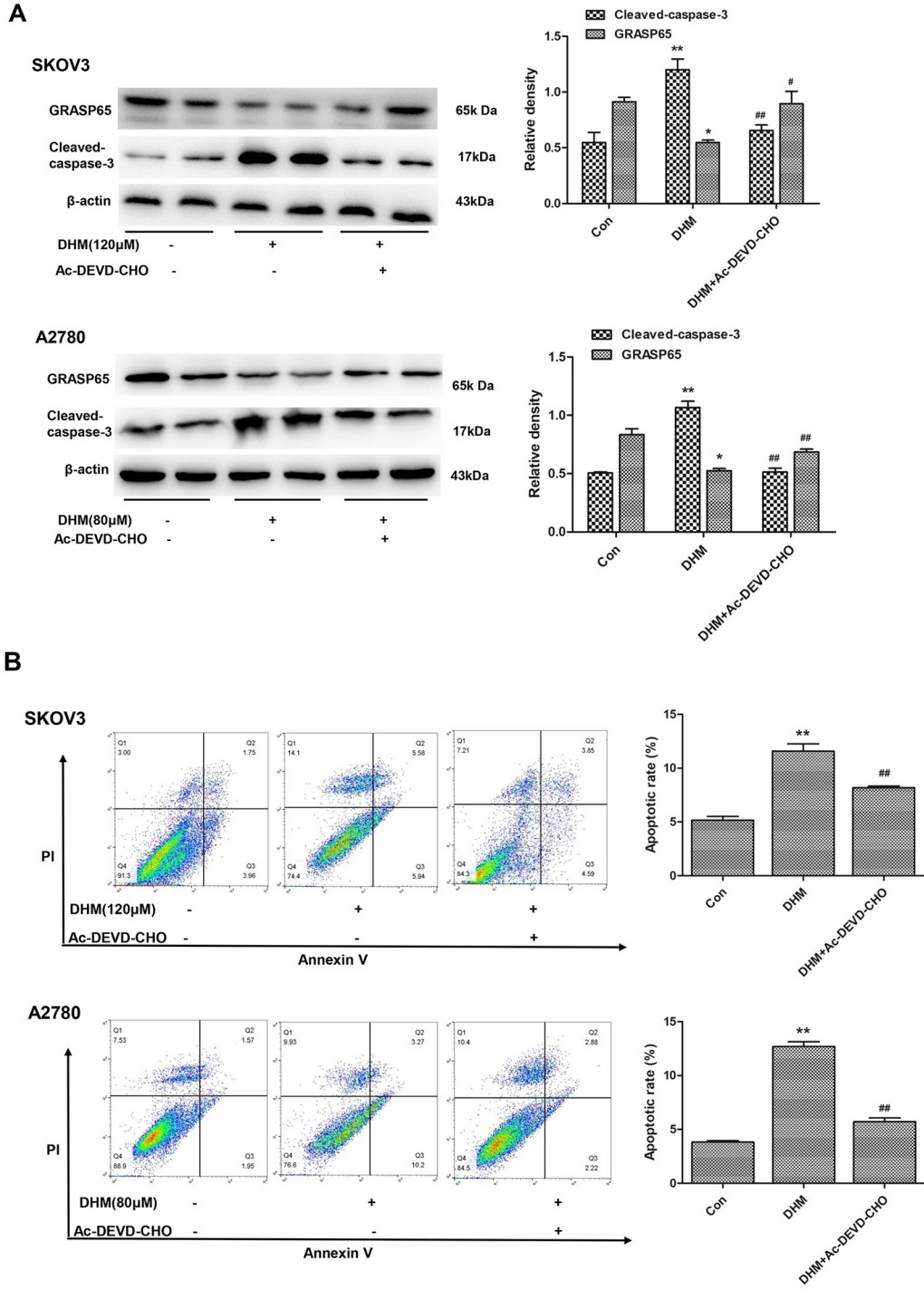

**Fig 4. Caspase-3 activation is crucial for suppression of GRASP65 expression in SKOV3 and A2780 cells. A**. Western blot analysis showed that the caspase-3 inhibitor Ac-DEVD-CHO dramatically attenuated DHM-induced effects in SKOV3 and A2780 cells. **B**. SKOV3 and A2780 cells were treated with 120 and 80 μM DHM for 48 h with or without pretreatment with Ac-DEVD-CHO for 30 min and analyzed by flow cytometry after Annexin V-FITC/PI staining. Annexin-V-FITC−/PI− populations in Q4 were living cells, while Annexin-V-FITC+/PI−cells in Q3 were undergoing necrosis, and Annexin-V-FITC+/PI+ cells in Q2 were either in the end stage of apoptosis or were already dead. The total populations in Q2 and Q3 were considered as apoptotic cells. Quantitative analysis of total apoptotic cells is shown. Each experiment was repeated at least three times. $^{*}$ $p < 0.05$, $^{**}p < 0.01$ vs the control group. $^{\#}p < 0.05$, $^{\#\#}p < 0.01$ vs the DHM group.

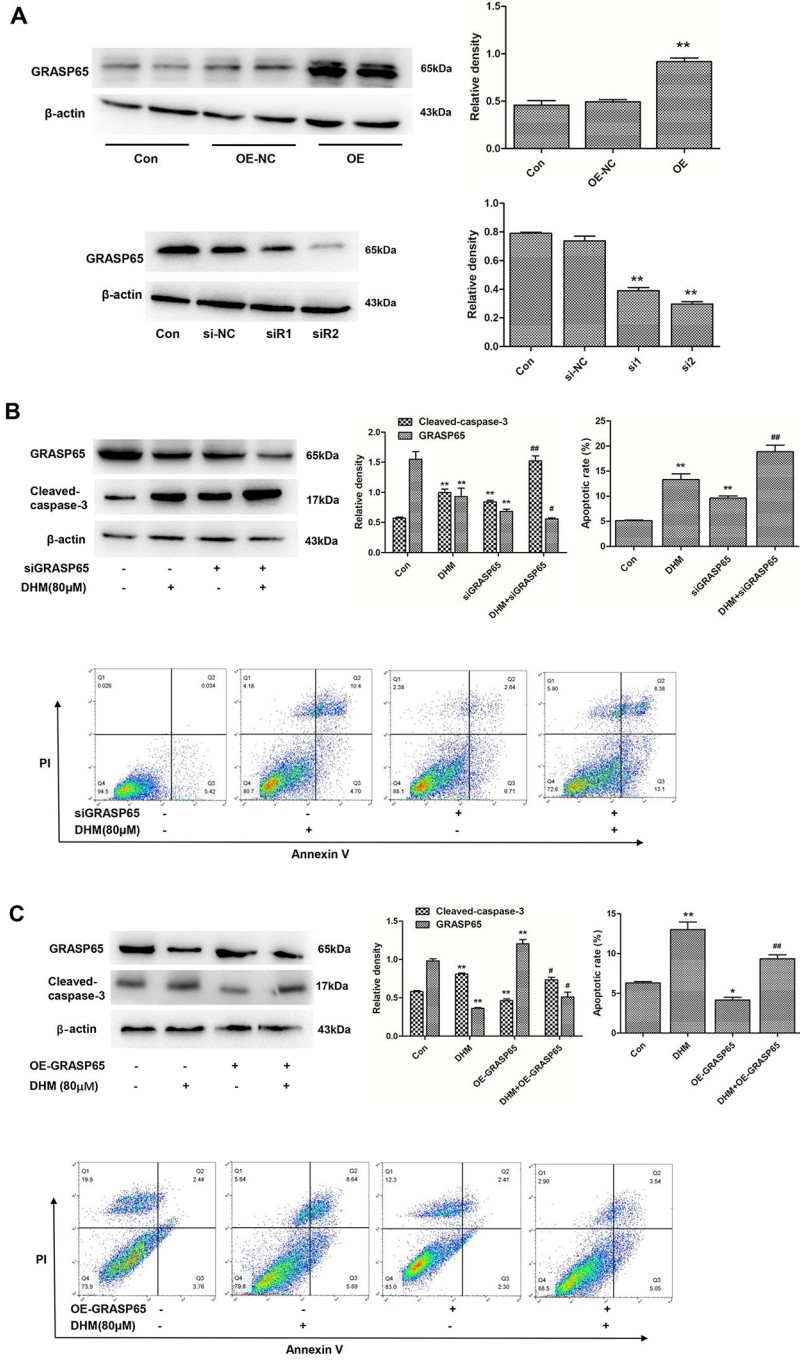

**Fig 5. Effects of GRASP65 on DHM-induced apoptosis in A2780 cells. A**. Transient transfection of *GRASP65* siRNA and overexpression plasmid was confirmed by western blot. **B/C**. The expression of GRASP65and cleaved-caspase-3 were assessed by western blotting, and apoptosis was determined using flow cytometry after transfection following treatment with 80 μM of DHM for 48 h. Quantitative analysis of total apoptotic cells is shown. *GRASP65* siRNA transfection results are shown in 5B, and overexpression plasmid is shown in 5C. Each experiment was repeated at least three times. * p < 0.05, ** p < 0.01 vs the control group. # p < 0.05, ## p < 0.01 vs the DHM group.

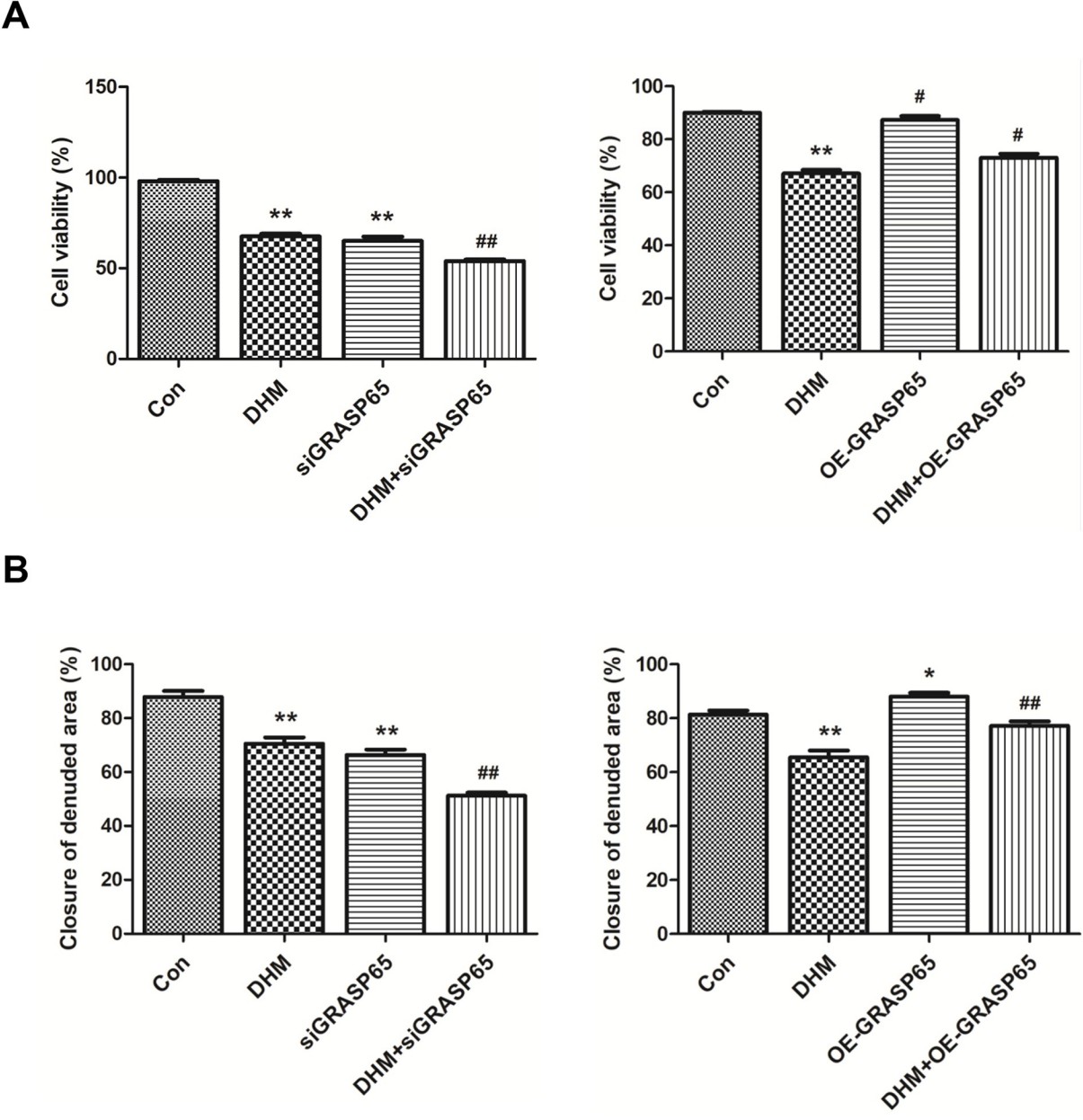

**Fig 6. Effects of GRASP65 on DHM-induced cell viability and migration in A2780 cells. A**. Cell viability was tested by CCK8 assay and **B**. cell migration was tested by wound healing assay after transfection following treatment with 80 μM of DHM for 48 h. Statistical analysis of cell viability and migration is shown. Each experiment was repeated at least three times. * $p < 0.05$, ** $p < 0.01$ vs the control group. # $p < 0.05$, ## $p < 0.01$ vs the DHM group.

transfection in A2780 cells. As shown in Fig 6A, the cell viability of A2780 cells transfected with *GRASP65* siRNA was lower than that in the control group with or without treatment with 80 μM of DHM. In comparison to the DHM-treated group, DHM suppressed wound closure in A2780 cells after transfected with *GRASP65* siRNA (Fig 6B).

On the contrary, overexpression of *GRASP65* could improve DHM-induced inhibition of cell viability (Fig 6A) and induce cell migration when treated with 80 μM of DHM (Fig 6B).

These results suggested that GRASP65 depletion using siRNA combined with DHM treatment had an additive effect on DHM-induced inhibition of cell viability and cell migration.

### JNK/ERK pathway participated in DHM-mediated apoptosis in A2780 cells

Previous studies have shown that the MAPK signaling pathway plays an important role in chemotherapy-induced apoptosis [37]. Three major MAPKs, namely ERK, JNK, and p38 MAPK, are activated by various stresses, including reactive oxygen species (ROS) [38] and can influence apoptosis either positively or negatively. Therefore, we first determined whether MAPKs were activated in DHM-treated A2780 cells. Western blot analysis showed that DHM induced activation of JNK and ERK in a dose-dependent manner, but no obvious changes in p38 level (Fig 7A).

We also found that there was no significant difference in p-JNK/ERK levels between the siGRASP65 group and the control group (Fig 7B), additionally, no difference between the DHM group and the DHM+siGRASP65 group (Fig 7B).

These results suggested that ERK/JNK signaling pathway involved in DHM-mediated cell apoptosis in A2780 cells, however, GRASP65 depletion had no effects on the p-JNK/ERK levels in A2780 cells.

### Suppression of the ERK/JNK pathway attenuated DHM-induced apoptosis in A2780 cells

To make the mechanism involved in inhibiting GRASP65 further clear, protein expression of GRASP65 and apoptosis-related proteins in A2780 cells were examined with the use of JNK and ERK inhibitors, respectively.

A2780 cells were pre-treated with 20 μM SP600125 (a JNK inhibitor) or 10 μM U0126 (an ERK inhibitor) for 2 h, followed by 80 μM DHM treatment for another 48 h and then were analyzed by western blot. Compared with the Control group, DHM increased the p-JNK/ERK levels and the caspase-3 cleavage and downregulated the expression of GRASP65 simultaneously. Interestingly, in comparison to the DHM group, SP600125 and U0126 significantly inhibited DHM-induced caspase-3 activation and increased GRASP65 levels after treatment with 80 μM DHM, followed by the inhibition of p-JNK/ERK levels (Fig 8A and 8B).

Taken together, these findings showed that the JNK/ERK pathway might be involved in DHM-mediated caspase-3 activation and GRASP65 inhibition in A2780 cells and suppression the ERK/JNK pathway could attenuate DHM-induced apoptosis, followed by an increase of GRASP65 expression.

## Discussion

The Golgi complex has been demonstrated to undergo fragmentation during apoptosis in several cancers. Due to the role of ER and Golgi during induction/execution of apoptosis, Golgi proteins have garnered significant interest as novel targets for selective anti-cancer therapies [3]. Cleavage of GRASP65 by caspase-3 correlates with Golgi fragmentation [7], and the fragmentation partially prevented by the expression of a caspase-resistant form of GRASP65 during apoptosis [29, 39]. GRASP65 also seems to be the important target of signaling events leading to Golgi breakdown during apoptosis [35]. Furthermore, GRASP65 and GRASP55 have been recently used as tools to disrupt the Golgi structure and thereby determine the functional consequence of Golgi structural disruption [40], which prompt numerous researchers to explore the underlying mechanisms of Golgi structure formation and function. Additionally, the Golgi-localized caspase-2 and -3 are generally accepted as central players in the execution phase of apoptosis, as they mediate cleavage of several golgins and GRASPs, including GM130

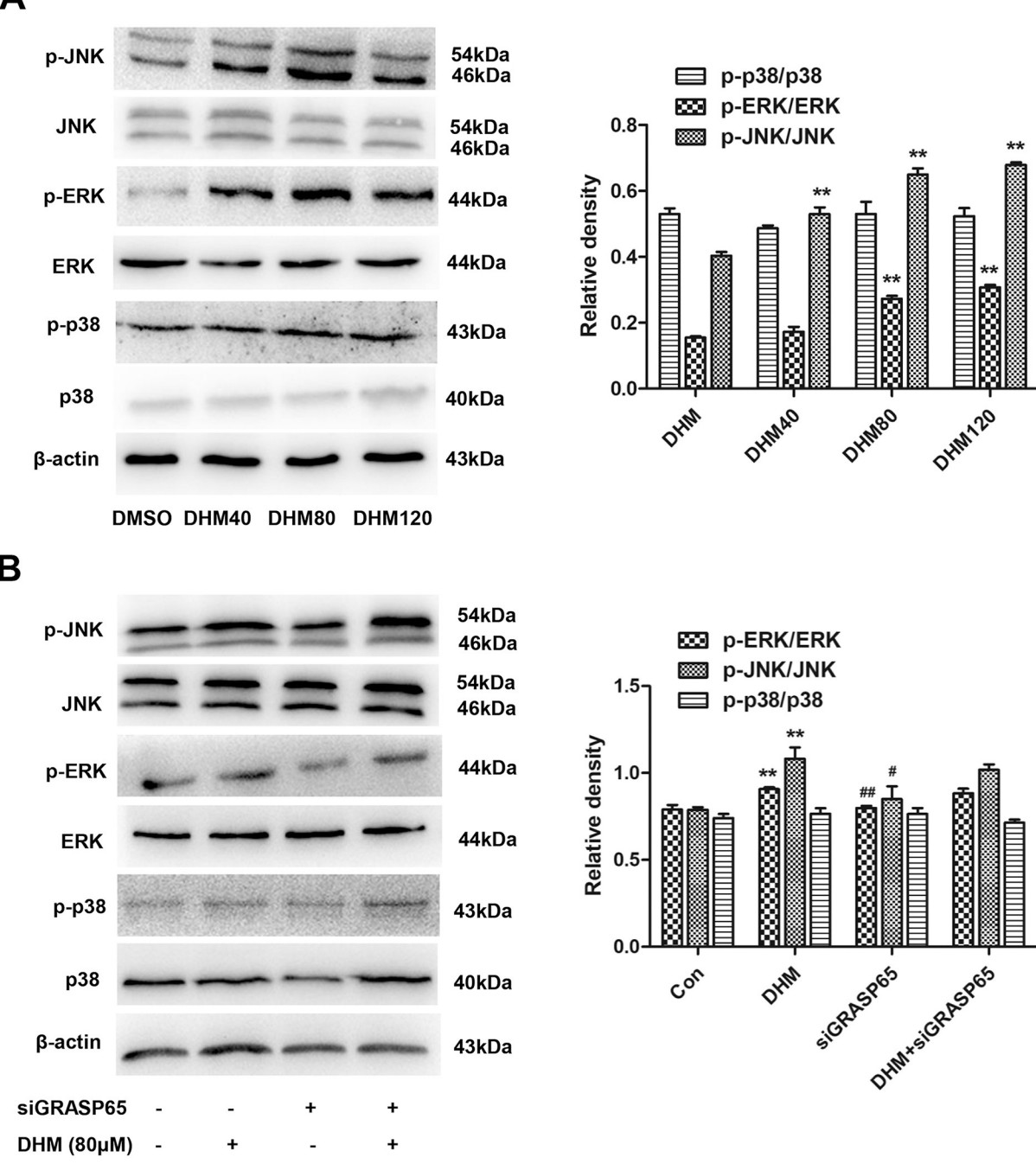

**Fig 7. ERK/JNK pathway participated in DHM-mediated apoptosis in A2780 cells. A**. A2780 cells were treated with various concentrations of DHM for 48 h, and the expressions of p-JNK/JNK, p-p38/p38MAPK and p-ERK/ERK were evaluated using western blot. **B**. p-JNK, p-p38 and p-ERK levels were detected by western blot after siGRASP65 transfection, following treatment with 80 μM of DHM for 48 h. Each experiment was repeated at least three times. * p < 0.05, **p < 0.01 vs the control group. #p < 0.05, ##p < 0.01 vs the DHM group.

[41] and GRASP65 [29]. Therefore, these studies indicated that Golgi proteins are potential therapeutic targets, as Golgi disruptive agents may facilitate Golgi fragmentation and induce apoptosis. In the present study, we mainly focused on the potential effects of GRASP65 in DHM-mediated ovarian cancer cell apoptosis.

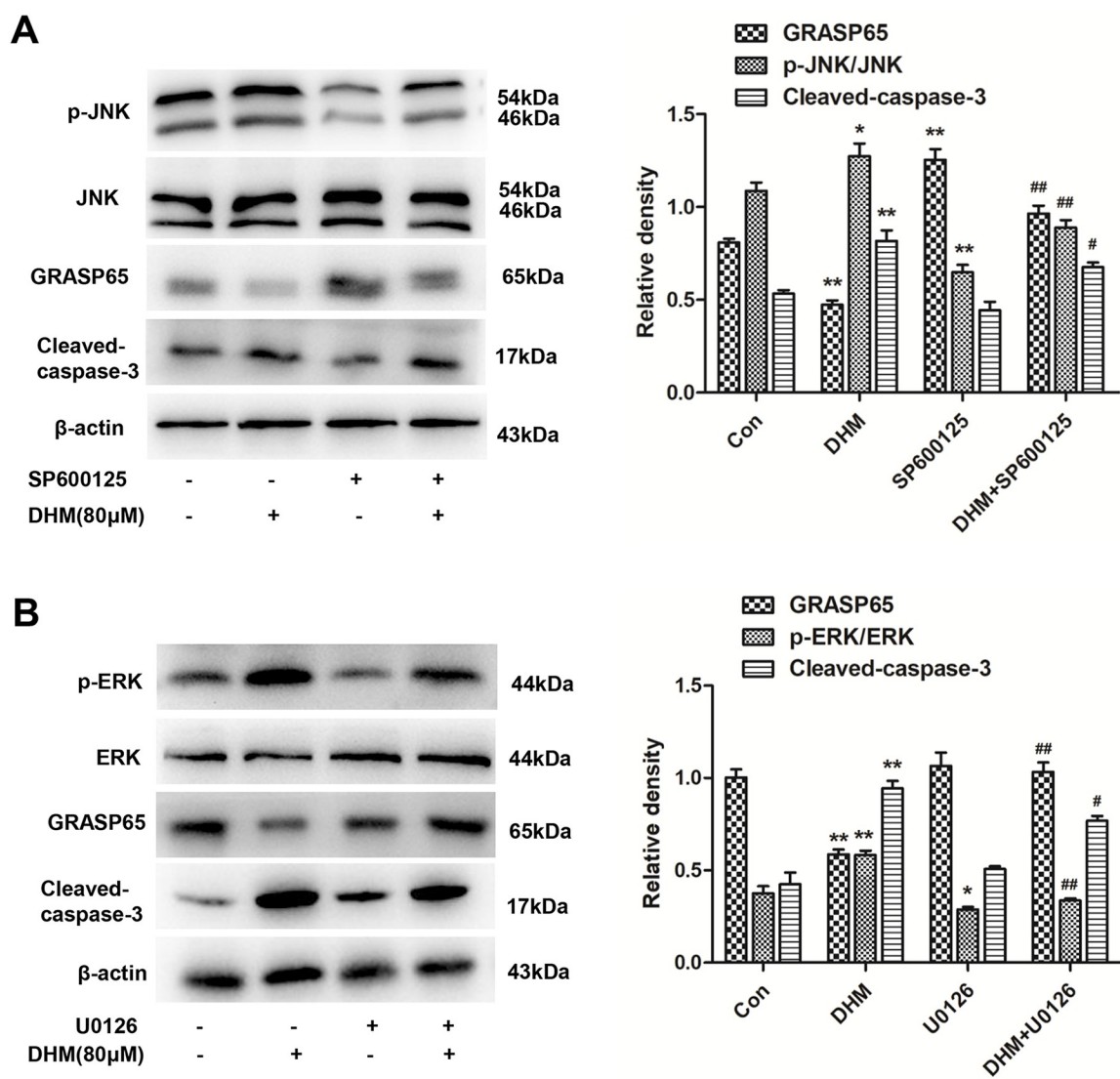

**Fig 8. Suppression of the ERK/JNK pathway attenuated DHM-induced apoptosis in A2780 Cells. A/B**. GRASP65, p-JNK/ERK and cleaved-caspase-3 levels were analyzed by western blot, following treatment with 80 μM of DHM in combination with SP600125 (8A) and U0126 (8B). Each experiment was repeated at least three times. * $p < 0.05$, ** $p < 0.01$ vs the control group. # $p < 0.05$, ## $p < 0.01$ vs the DHM group.

DHM, a natural flavonoid derived from *Ampelopsis grossedentata*, has received considerable interest as a potential candidate for cancer therapy. Many studies have demonstrated that DHM has functions in cancer prevention and development [21–23]. Interestingly, an inverse relationship between the intake of dietary flavonoid and cancer risk has been observed in different studies [42]. Therefore, we sought to further investigate the antitumor activity of DHM. Cell apoptosis is an active process of endogenous programmed cell death, which is a standard benchmark for the selection of anticancer drugs [43], and its deregulation is a fundamental hallmark of cancer development and progression [44]. The results of our study showed that DHM treatment significantly promoted cell apoptosis by upregulating the proportion of Bax/ Bcl-2 and activating caspase-3 in SKOV3 and A2780 cells. Similarly, many chemotherapies,

such as cisplatin and paclitaxel, aim to cure or control cancer by inducing apoptosis of human carcinoma cells [45].

To further determine the effects of GRASP65 on DHM-induced cell apoptosis, we first examine GRASP65 expression using immunoblotting analysis. GRASP65 is a specific substrate of caspase-3[36], and cleavage of GRASP65 correlates with Golgi fragmentation, which can be inhibited by the expression of a caspase-resistant form of GRASP65[29]. And caspase cleavage of Golgi structural proteins may be the downstream result of effector caspase activation, allowing the packaging of Golgi remnants into apoptotic blebs for disposal [29]. Our findings showed that GRASP65 expression decreased during DHM-induced apoptosis, likely due to caspase-3 cleavage according to the reports that showed caspase-3 cleavage of GRASP65 is necessary for apoptotic Golgi fragmentation [29,36]. Furthermore, we found inhibition of caspase-3 activity by Ac-DEVD-CHO could mitigate DHM-induced cell apoptosis to delay or reduce cell death, followed by an increase of GRASP65 level. Therefore, we speculated that DHM might activate caspase-3, which is closely related to the suppression of GRASP65 expression during DHM-induced apoptosis. However, the significance of GRASP65 suppression and its relationship with DHM-mediated effects remain obscure.

Then, we performed *GRASP65* siRNA and overexpression plasmid transfections to further detect the potential role of GRASP65 in DHM-induced cell apoptosis. Previous research reported that pharmacological intervention or overexpression of the C-terminal fragment of GRASP65 inhibits GF and decreases or delays neuronal cell death [46]. Depletion of GRASP65 by siRNA reduced the number of cisternae in the Golgi stacks, which can be rescued by expressing exogenous GRASP65 [47]. In addition, depletion of GRASP65 reduced cell attachment and migration [48], even resulted in cell death [36,49]. Numerous researches have indicated that the Golgi apparatus and GA fragmentation play important roles in apoptosis [28, 50] and GA is a sensor and common downstream effector of stress signals in cell death pathways [51]. Similarly, flow cytometry and western blot analysis in our study showed that overexpression of *GRASP65* reduced cell apoptosis and inhibition of *GRASP65* by siRNA induced apoptosis in A2780 cells. On the other hand, overexpression of *GRASP65* mitigated DHM-mediated cell apoptosis. Conversely, GRASP65 depletion combined with DHM treatment could further promoted cell apoptosis, suggesting that DHM combined with GRASP65 intervention may elicit an antitumor response in ovarian cancer cells. Therefore, GRASP65 downregulation may have a critical role in DHM-induced apoptosis in OCs, while its role in DHM-induced Golgi morphological changes may be complex.

However, the molecular mechanism of DHM-mediated suppression of GRASP65 expression is unclear. Several studies have shown that the mitogen-activated protein kinase (MAPK) signaling pathway plays an important role in chemotherapy-induced apoptosis [36,52–53]. JNK can function as pro-apoptotic and anti-apoptotic kinases in different cell types [54]. Accumulating evidence suggests that phosphorylation of GRASP65 by kinases, such as ERK [55, 56], JNK2[57], Cdk1 and PLK1, or cleavage by caspase, is required for mitotic or apoptotic Golgi fragmentation [8,58]. JNK2–GRASP65 signaling has a prominent role in the identification of novel anti-cancer agents that block cell cycle progression [57]. Once activated, JNKs in turn phosphorylate several transcription factors and cellular proteins that are associated with apoptosis, including Bcl2[53], capase-3[59], and others. In the present study, DHM could activate JNK/ERK signaling pathway in a concentration-dependent manner. On the other hand, inhibition of JNK and ERK signaling suppressed the cleavage of caspase-3 as well as increased the expression of GRASP65 in A2780 cells treated with DHM, indicating that DHM might activate JNK/ERK-caspase-3 pathway and the activation of caspase-3 was crucial for the downregulation of GRASP65. Moreover, we found DHM combined with *GRASP65* siRNA intervention couldn't further affect the p-JNK and p-ERK levels, only induce apoptosis. Increasing

evidence suggests that the GA is a crucial downstream effector [51], even as a downstream target organelle of endoplasmic reticulum and mitochondria associated with GRASP65 phosphorylation when oxidative stress occurs [60,61]. Based on the findings, we speculated that GRASP65 suppression might be the downstream effector during DHM-mediated apoptosis in ovarian cancer cells, which was at least partially due to the activation of the JNK/ERK pathway.

Taken together, the present study demonstrated that DHM treatment promoted JNK/ERK activation and the cleavage of Caspase-3, which was crucial for the suppression of GRASP65 in ovarian cancer cells. Furthermore, DHM treatment combined with *GRASP65* depletion had an additive role in DHM-mediated anticancer effects. However, the role of GRASP65 in DHM-induced Golgi distribution and morphology remains unclear. In the long term, future studies employing techniques, such as electron microscopy and confocal scanning microscopy, are needed to further reveal the structure-function relationships of the GA in DHM-induced cell apoptosis.

## Supporting information

**S1 Fig.**
(TIF)

**S2 Fig.**
(TIF)

**S3 Fig.**
(TIF)

**S4 Fig.**
(TIF)

**S5 Fig.**
(TIF)

**S6 Fig.**
(TIF)

## Acknowledgments

We would like to thank LetPub (www.letpub.com) for providing linguistic assistance during the preparation of this manuscript.

## Author Contributions

**Conceptualization:** Xianbing Chen.

**Funding acquisition:** Depei Yuan.

**Investigation:** Depei Yuan, Yongfen Yi.

**Project administration:** Fengjie Wang, Yongfen Yi.

**Supervision:** Yi Luo.

**Writing – original draft:** Fengjie Wang.

**Writing – review & editing:** Yi Luo.

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
