## [Decision Letter · Decision Letter 0]

30 Aug 2019

PONE-D-19-21799

Golgi reassembly and stacking protein 65 downregulation is required for the anti-cancer effect of dihydromyricetin on human ovarian cancer cells

PLOS ONE

Dear Dr luo,

Thank you for submitting your manuscript to PLOS ONE. After careful consideration, we feel that it has merit but does not fully meet PLOS ONE’s publication criteria as it currently stands. Therefore, we invite you to submit a revised version of the manuscript that addresses the points raised during the review process.

We would appreciate receiving your revised manuscript by Oct 14 2019 11:59PM. To enhance the reproducibility of your results, we recommend that if applicable you deposit your laboratory protocols in protocols.io, where a protocol can be assigned its own identifier (DOI) such that it can be cited independently in the future. For instructions see: http://journals.plos.org/plosone/s/submission-guidelines#loc-laboratory-protocols

We look forward to receiving your revised manuscript.

Kind regards,

Yi-Hsien Hsieh, Ph.D.

Academic Editor

PLOS ONE

Journal Requirements:

Reviewers' comments:

Reviewer's Responses to Questions

**Comments to the Author**

1. Is the manuscript technically sound, and do the data support the conclusions?

Reviewer #1: Yes

Reviewer #2: Partly

Reviewer #3: Partly

2. Has the statistical analysis been performed appropriately and rigorously? 

Reviewer #1: Yes

Reviewer #2: Yes

Reviewer #3: I Don't Know

3. Have the authors made all data underlying the findings in their manuscript fully available?

Reviewer #1: Yes

Reviewer #2: Yes

Reviewer #3: Yes

4. Is the manuscript presented in an intelligible fashion and written in standard English?

Reviewer #1: Yes

Reviewer #2: Yes

Reviewer #3: Yes

5. Review Comments to the Author

Reviewer #1: the present article entitled: Golgi reassembly and stacking protein 65 downregulation is required for the anticancer effect of dihydromyricetin on human ovarian cancer cells, the authors focused to investigate the anti-tumor effects of DHM in OC cells and also elucidated the associated underlying molecular mechanisms. Overall, the manuscript was clearly written and the data was obviously presented. However, several points should be seriously taken in consideration for the following raisons:

1- In the discussion section, the authors should cite more references.

2- The authors should add n=? in each legend to figure.

3- Why the authors did not investigated the expression of cell cycle regulatory proteins such as CDK?

4- In the flow cytometry analysis of apoptotic cells figures 4B and 5C, the Y-axis must be moved to be on 102 for control, DHM and DHM+Ac-DEVD-CHO-treated cells.

Reviewer #2: This manuscript presents the results of studies examining whether DHM-induced antitumor activity was through the downregulation of GRASP65 in SKOV3 and A2780 cells. They found that the suppressive effects of proliferation, migration, and the promotion of apoptosis induced by DHM was regulated by GRASP65. Since research on the involvement of Goldi proteins with DHM-induced antitumor effects has not been done so far, the topic is of interest. However, the studies are not sufficiently verified.

1. Although authors explained why they focused on the involvement of GRASP65 in DNM-induced anti-tumor activity, it was difficult to understand. Authors should explain how GRASP65 is regulated by ERK, CDK1, and PLK-1 specifically and whether the regulation is about the expression or morphological changes.

2. Fig. 4 showed that DHM-induced caspase-3 activation was crucial for suppression of GRASP65 expression and induction of apoptosis by DHM. This result did not imply that activated caspase-3 mediated cleavage and reduction of GRASP65 was crucial for DHM-induced cell apoptosis. The caption [GRASP65 was essential for the anti-cancer effects of DHM in A2780 cells] was not correct.

3. In Fig. 5, the results of western blotting and its quantitative results did not match. For example, in Fig. 5A, the inductive effect of OE2 was not obvious by western blotting, but the 1.5-fold inductive effect was observed by a densitometric analysis. In Fig. 5B, the combination effect of siGRASP65 and DHM on GRASP65 expression was not obvious by western blotting, but the additive effect was shown by a densitometric analysis.

4. In Fig. 5, both siGRASP65 and OE-GRASP65 increased the number of apoptotic cells. However, the combination of DHM and OE-GRASP65 attenuated the DHM-induced apoptotic effects. Is this correct?

5. In Fig. 6, authors should examine the effects of OE-GRASP65 on cell viability and migration. Furthermore, the result of invasion should be included.

6. In Fig. 7, the results of western blotting and its quantitative results did not match. Authors showed p38 level did not change, but it looks like that DHM suppressed the phosphorylation of p38 by western blotting. Furthermore, the addictive effects of siGRASP65 plus DHM on the expression of p-JNK and p-ERK were not observed.

7. As compared with the results of Fig. 7, the inductive effects on the expression of p-JNK and p-ERK by DHM were weak.

Reviewer #3: In this manuscript, Wang et al. aimed to identify a functional connection between the antitumor activity of Dihydromyricetin (DHM) and Golgi reassembly-stacking protein of 65 kDa (GRASP65) through a mechanism that involves activation of apoptosis in ovarian cancer cell lines. DHM is a flavonoid found in several species, and it has proapoptotic activity on several cancer cell lines, including of hepatoma, melanoma, osteosarcoma, gastric cancer, and ovarian cancer. GRASP65 is a Golgi apparatus protein implicated in several aspects of protein trafficking and in the structure of the Golgi apparatus including the stacking of Golgi cisternae and the linking of Golgi stacks to form a Golgi ribbon. In addition to confirming published data indicating that DHM has proapoptotic activity on ovarian cancer cells, this manuscript provides evidence that it also negatively affects cell migration and invasion of these cells. As for the proapoptotic activity, the data indicate that DHM-induced apoptosis proceeds with an increase in the levels of cleaved caspase-3, which correlates with a decrease in the levels of GRASP65. This is an expected correlation, because it is well known that GRASP65 is a target of activated caspase-3. The manuscript present data of GRASP65 RNAi and overexpression experiments designed to demonstrate the causality in the DHM-induced apoptosis that this Golgi protein might be involved in. Finally, the authors explore the signaling pathways that might be implicated in the apoptotic response to DHM. Overall, the manuscript shows data of good standard. However, the major conclusion that GRASP65 is required for the anti-cancer effect of DHM is not supported by the data provided. In addition, a number of major and minor issues do not warrant publication of the manuscript as it is.

Major issues:

1) As mentioned, the main conclusion of the manuscript that GRASP65 is required for the anti-cancer effect of DHM has no experimental support. Moreover, the results of the GRASP65 RNAi experiments show the opposite, which is that the treatment with DHM in cells knocked down of GRASP65 expression resulted in increased levels of cleaved caspase-3 and apoptotic rate, compared to control cells or to each of the individual treatments. If any, the conclusion here is that the reduction in the levels of GRASP65 cooperates in the proapoptotic effect of DHM, or vice versa. In fact, the data show that the knocking down of GRASP65 has also a proapototic effect, and that the treatment with DHM has an additive proapoptotic effect. Considering the results presented, the testing of the requirement of GRASP65 function for the proapoptotic activity of DHM needs a different experimental design.

2) An intriguing set of results is that the overexpression of GRASP65 also resulted in increased levels of cleaved caspase-3 and apoptotic rate, but somehow combined with DHM resulted in less proapoptotic effect compared to the treatment with DHM alone. The authors should comment on these results providing a possible explanation.

Minor issues:

3) The revised version should contain line numbering, as it is an editorial request, otherwise the revision process is time consuming.

4) Although overall the manuscript is read and understood, it is advised an additional professional scientific proofreading as some of the statements are incorrect and thus are potentially misleading.

5) In the "Abstract" the statement "...DHM inhibited cell migration..." is incorrect; DHM reduced cell migration.

6) In "Introduction" the statement "...leads to depolymerization and division of the Golgi..." is incorrect; the Golgi is not a polymer and it does not divide.

7) The sequence of each GRASP65 siRNA oligonucleotide should be provided, as well as a description of each of the plasmids used for the overexpression of GRASP65.

8) The results shown in Fig 1A should be removed, because similar results are already published.

9) In "Results" the statement "...and almost completely blocked the closure..." is incorrect; at the most, the treatments reduced SKOV3 migration or invasion to ∼25% the respective levels observed in control conditions.

10) The title of Fig. 1 legend is incorrect, because DHM did not inhibit cell viability, cell migration and cell invasion; it reduced the extent of these processes.

11) In "Results" the statement "...DHM downregulated the expression of GRASP65 in a concentration-dependent manner, followed by activation of Caspase-3"... is odd. The published data indicate that the process is the opposite, meaning that during apoptosis the activation of caspase-3 results in cleavage of GRASP65 and thus in GRASP65 downregulation.

12) In the legend of Fig. 3 " #p < 0.05, ##p < 0.01 vs the DHM group" should be removed.

13) In "Results" the title "GRASP65 was essential for the anti-cancer effects of DHM in A2780 cells" is incorrect, because the data does not show at all that GRASP65 is essential for the effects of DHM.

14) In the same section of "Results" the statement "... cells were pre-treated with a specific caspase-3 inhibitor, Ac-DEVD-CHO, for 30 min to suppress the effects of DHM..." is odd, because the experiment should not have been designed to suppress the effects of DHM, but instead to evaluate the contribution of caspase-3 in the effects of DHM.

15) The title of Fig. 4 legend is odd; please revise.

16) In "Results" the statement "These results suggested that downregulation of GRASP65 could promote DHM-induced inhibition of cell viability and cell migration" is at least speculative, and should be revised. The data indicate that the effects of GRASP65 RNAi and DHM treatment are additive and thus very unlikely to be mechanistically related.

17) Please revise the magnitude informed of the scale bar in the legend of Fig. 6 as it seems very similar to that of Figures 2 and 3.

18) Please, provide a rationale for not performing all the subsequent analyses in SKOV3 cells.

19) Please, explain why it was not performed the analysis of the levels of p-p38 in cells transfected with siGRASP65.

20) In "Discussion" the statement "The Golgi is essential for the endoplasmic reticulum and mitochondria..." is odd; please revise.

21) In "Discussion" the statement "GRASP65, a peripheral Golgi membrane protein, is required for mitotic or apoptotic Golgi fragmentation when specifically cleaved by caspases" is odd; please revise.

22) In "Discussion" the statement "...Golgi execution phase of apoptosis..." is odd; please revise.

23) In "Discussion" the statement "This implies that the Golgi is a potential therapeutic target, as Golgi disruptive agents may facilitate Golgi fragmentation and induce apoptosis" is a hypothesis already tested by several groups with several published examples in the literature. The authors should discuss their findings in the context of the published data.

24) In "Discussion" the statement "...Golgi formation may be carcinogenic, or a consequence of cancer progression" is immensely odd; please provide more explanations or revise.

25) In "Discussion" the statement "... inhibiting cleaved caspase-3 can block apoptotic cell death and increasing Caspase-3-like protease activity may be responsible for the delayed cell death" is odd; please revise.

26) In "Discussion" the statement "... DHM may activate caspase-3, which then cleaves and reduces GRASP65 expression to promote cell apoptosis" is an overstatement, because the data do not support the conclusion that the proapoptotic effects of DHM are mediated by the reduction in the levels of GRASP65.

27) In "Discussion" the statement "... activated caspase-3-mediated cleavage and the reduction of GRASP65 was crucial for DHM-induced cell apoptosis" is redundant, and, again, is not supported by the data.

28) The complete "Conclusion" section should be revised, because many statements are misleading (considering that some conclusions are incorrect).

29) In Fig. 3A and 3B the expected effect of DHM is the fragmentation of the Golgi apparatus. Because DHM resulted in a decrease in the levels of GRASP65, a different Golgi resident protein should be analyzed by immunofluorescence. Also, to diagnose Golgi apparatus fragmentation (instead of Golgi vesiculation) simultaneous immunofluorescence of at least cis and trans Golgi resident proteins should be provided. Higher magnification of the Golgi ribbon in control-treated cells and DHM-treated cells should be also included to properly assess Golgi fragmentation.

6. PLOS authors have the option to publish the peer review history of their article (what does this mean?). If published, this will include your full peer review and any attached files.

Reviewer #1: No

Reviewer #2: No

Reviewer #3: No

---

## [Author Response · Author response to Decision Letter 0]

8 Oct 2019

Dear Editor and Reviewers:

Thank the editor and the reviewers for your positive comments and constructive suggestions for our manuscript entitled “Golgi reassembly and stacking protein 65 downregulation is required for the anti-cancer effect of dihydromyricetin on human ovarian cancer cells” (Manuscript ID: PONE-D-19-21799). Your positive comments and useful suggestions that are beneficial for the further improvement of this manuscript are highly appreciated. Based on the reviewer’s suggestions, we have made double check and careful modifications for this manuscript. The modified or revised parts in the revised manuscript are shown in tracking changes. 

We’re sure that our manuscript meets PLOS ONE's style requirements and have provided the original uncropped and unadjusted images underlying all blot or gel results reported in Supporting Information files. 

Meanwhile, the point-to-point responses to address the concerns of the reviewers are listed as follows: 

Review Comments to the Author

Reviewer #1: the present article entitled: Golgi reassembly and stacking protein 65 downregulation is required for the anticancer effect of dihydromyricetin on human ovarian cancer cells, the authors focused to investigate the anti-tumor effects of DHM in OC cells and also elucidated the associated underlying molecular mechanisms. Overall, the manuscript was clearly written and the data was obviously presented. However, several points should be seriously taken in consideration for the following raisons:

1- In the discussion section, the authors should cite more references. 

Response: Thank the reviewer for giving this useful suggestion. We have added and cited more references and further discussed the issue in the discussion section as shown in the revised manuscript. 

2- The authors should add n=? in each legend to figure. 

Response: Thank the reviewer for pointing out the issue. We have added the sentence as “Each experiment was repeated at least three times” in each legend to figures according to the suggestion. 

3- Why the authors did not investigated the expression of cell cycle regulatory proteins such as CDK? 

Response: We would like to express our highest appreciation to the reviewer for this useful suggestion. The effects of DHM on the expression of cell cycle regulatory proteins, such as CDK, haven’t been detected, then we would like to further examine and provide the results in the future research. 

4- In the flow cytometry analysis of apoptotic cells figures 4B and 5C, the Y-axis must be moved to be on 102 for control, DHM and DHM+Ac-DEVD-CHO-treated cells. 

Response: Thank the reviewer for pointing out the issue. We have modified and re-analyzed the results of flow cytometry in the revised manuscript. 

Reviewer #2: This manuscript presents the results of studies examining whether DHM-induced antitumor activity was through the downregulation of GRASP65 in SKOV3 and A2780 cells. They found that the suppressive effects of proliferation, migration, and the promotion of apoptosis induced by DHM was regulated by GRASP65. Since research on the involvement of Goldi proteins with DHM-induced antitumor effects has not been done so far, the topic is of interest. However, the studies are not sufficiently verified.

1. Although authors explained why they focused on the involvement of GRASP65 in DNM-induced anti-tumor activity, it was difficult to understand. Authors should explain how GRASP65 is regulated by ERK, CDK1, and PLK-1 specifically and whether the regulation is about the expression or morphological changes. 

Response: We really appreciate the reviewer to point out the important issue. Most studies on GRASP65 were focused on the regulation of morphological changes by ERK, CDK1, and PLK-1. And we found DHM could affect the expression of GRASP65, we investigated the involvement and function of GRASP65 in DNM-induced anti-tumor activity firstly. Meanwhile, we want to further explore the morphological changes of GA and the relation between the morphology and function in DHM-induced effects in OCs in the future research. 

2. Fig. 4 showed that DHM-induced caspase-3 activation was crucial for suppression of GRASP65 expression and induction of apoptosis by DHM. This result did not imply that activated caspase-3 mediated cleavage and reduction of GRASP65 was crucial for DHM-induced cell apoptosis. The caption [GRASP65 was essential for the anti-cancer effects of DHM in A2780 cells] was not correct. 

Response: Thank the reviewer for pointing out the issue. We have modified the caption as “DHM-induced caspase-3 activation was crucial for suppression of GRASP65 expression” for Fig.4 in the revised manuscript and further discussed the results again. 

3. In Fig. 5, the results of western blotting and its quantitative results did not match. For example, in Fig. 5A, the inductive effect of OE2 was not obvious by western blotting, but the 1.5-fold inductive effect was observed by a densitometric analysis. In Fig. 5B, the combination effect of siGRASP65 and DHM on GRASP65 expression was not obvious by western blotting, but the additive effect was shown by a densitometric analysis. 

Response: Thank the reviewer for pointing out the issue. We have exchanged the relative figure of western blot analysis and re-analyzed the results. Each experiment was repeated at least three times. Therefore, we’re sure that the effects of siRNAs to suppress GRASP65 and OE plasmid to overexpress GRASP65 were obvious by a densitometric analysis as shown in Fig.5 in the revised manuscript. 

4. In Fig. 5, both siGRASP65 and OE-GRASP65 increased the number of apoptotic cells. However, the combination of DHM and OE-GRASP65 attenuated the DHM-induced apoptotic effects. Is this correct? 

Response: We really appreciate the reviewer for pointing out this issue. We have performed the experiment again in this part, then we found that OE-GRASP65 plasmid transfection could reduce cell apoptosis. In addition, compared with DHM treatment group, combination of DHM treatment and OE-GRASP65 attenuated the DHM-induced apoptosis. Therefore, we re-analyzed the statistic results of flow cytometry as shown in Fig.5 in the revised manuscript.

5. In Fig. 6, authors should examine the effects of OE-GRASP65 on cell viability and migration. Furthermore, the result of invasion should be included. 

Response: Thank the reviewer giving us this useful suggestion. We have added the detection of the effects of OE-GRASP65 on cell viability and migration according to the reviewer’s suggestion as shown in Fig.6 in the revised manuscript. However, we only performed the test of cell invasion using Transwell assay in the previous study to test DHM-mediated effects for some reasons. We would like to further detect the effects of siGRASP65 and OE-GRASP65 on cell invasion and provide better results in the following study. 

6. In Fig. 7, the results of western blotting and its quantitative results did not match. Authors showed p38 level did not change, but it looks like that DHM suppressed the phosphorylation of p38 by western blotting. Furthermore, the addictive effects of siGRASP65 plus DHM on the expression of p-JNK and p-ERK were not observed.

Response: Thank the reviewer give us the suggestion for modifying this confusing result. Firstly, we’re sure that there was no obvious difference in the phosphorylation of p38 levels among the DHM treated groups based on the results, then we provided better results and figures of western blotting analysis as shown in Fig.7 in the revised manuscript.

In addition, according to the reviewer’s suggestion, we have re-performed the experiment to examine the effects of siGRASP65 plus DHM on the expression of p-JNK and p-ERK and found there was no obviously additive effects. Simultaneously, we speculated that GRASP65 downregulation might be the downstream effector in DHM-mediated activation of JNK/ERK-caspase-3 pathway according to the relative reference. But it is needed to confirm the relation in the following study. 

7. As compared with the results of Fig. 7, the inductive effects on the expression of p-JNK and p-ERK by DHM were weak. 

Response: Thank the reviewer give us the suggestion for modifying this confusing result. In fact, we have repeated the experiments and analyzed the results again according to your suggestion. Then we found that there was no obvious difference in p-JNK/ERK levels between the DHM group and DHM+siGRASP65 group, suggesting that GRASP65 depletion had no effects on the p-JNK/ERK levels. 

Therefore, we have modified them as shown in Fig.7 in the revised manuscript. 

Reviewer #3: In this manuscript, Wang et al. aimed to identify a functional connection between the antitumor activity of Dihydromyricetin (DHM) and Golgi reassembly-stacking protein of 65 kDa (GRASP65) through a mechanism that involves activation of apoptosis in ovarian cancer cell lines. DHM is a flavonoid found in several species, and it has proapoptotic activity on several cancer cell lines, including of hepatoma, melanoma, osteosarcoma, gastric cancer, and ovarian cancer. GRASP65 is a Golgi apparatus protein implicated in several aspects of protein trafficking and in the structure of the Golgi apparatus including the stacking of Golgi cisternae and the linking of Golgi stacks to form a Golgi ribbon. In addition to confirming published data indicating that DHM has proapoptotic activity on ovarian cancer cells, this manuscript provides evidence that it also negatively affects cell migration and invasion of these cells. As for the proapoptotic activity, the data indicate that DHM-induced apoptosis proceeds with an increase in the levels of cleaved caspase-3, which correlates with a decrease in the levels of GRASP65. This is an expected correlation, because it is well known that GRASP65 is a target of activated caspase-3. The manuscript present data of GRASP65 RNAi and overexpression experiments designed to demonstrate the causality in the DHM-induced apoptosis that this Golgi protein might be involved in. Finally, the authors explore the signaling pathways that might be implicated in the apoptotic response to DHM. Overall, the manuscript shows data of good standard. However, the major conclusion that GRASP65 is required for the anti-cancer effect of DHM is not supported by the data provided. In addition, a number of major and minor issues do not warrant publication of the manuscript as it is.

Major issues:

1) As mentioned, the main conclusion of the manuscript that GRASP65 is required for the anti-cancer effect of DHM has no experimental support. Moreover, the results of the GRASP65 RNAi experiments show the opposite, which is that the treatment with DHM in cells knocked down of GRASP65 expression resulted in increased levels of cleaved caspase-3 and apoptotic rate, compared to control cells or to each of the individual treatments. If any, the conclusion here is that the reduction in the levels of GRASP65 cooperates in the proapoptotic effect of DHM, or vice versa. In fact, the data show that the knocking down of GRASP65 has also a proapototic effect, and that the treatment with DHM has an additive proapoptotic effect. Considering the results presented, the testing of the requirement of GRASP65 function for the proapoptotic activity of DHM needs a different experimental design. 

Response: We would like to express our highest appreciation to the reviewer for pointing out the confusing conclusion. Frankly speaking, we might not provide a proper description to the present results in the manuscript. According to the reviewer’s suggestion, we couldn’t confirm the main conclusion of the manuscript based on the present results and we would like to further design the experiment comprehensively. However, considering the results presented, we could confirm that GRASP65 depletion has a proapoptotic effect and siGRASP65 combined with DHM treatment has an additive proapoptotic effect in ovarian cancer cells. We have modified the results in the revised manuscript. 

We would like to receive your useful and constructive suggestion again. 

2) An intriguing set of results is that the overexpression of GRASP65 also resulted in increased levels of cleaved caspase-3 and apoptotic rate, but somehow combined with DHM resulted in less proapoptotic effect compared to the treatment with DHM alone. The authors should comment on these results providing a possible explanation. 

Response: Thank the reviewer give us the suggestion for modifying this confusing result. In fact, the experiment data provided previously are misleading. We have repeated the experiment according to the suggestion and modified the results. The present results showed that overexpression of GRASP65 reduced cell apoptosis with the use of western blotting and FCM analysis and overexpression GRASP65 combined with DHM attenuated the proapoptotic effect compared to the DHM treatment alone. This is consistent with many reports about the effects of overexpression of GRASPs. We have modified the results and discussion sections as shown in the revised manuscript. 

Minor issues: 

3) The revised version should contain line numbering, as it is an editorial request, otherwise the revision process is time consuming. 

Response: Thank the reviewer for the useful suggestion. We have added the line numbering in the revised manuscript. 

4) Although overall the manuscript is read and understood, it is advised an additional professional scientific proofreading as some of the statements are incorrect and thus are potentially misleading. 

Response: Thank the reviewer for pointing out the issue. We have made double check and careful modifications for this manuscript and got linguistic assistance from LetPub. 

5) In the "Abstract" the statement "...DHM inhibited cell migration..." is incorrect; DHM reduced cell migration. 

Response: Thank the reviewer for pointing out the issue. We have modified the statement as “The present study showed that DHM reduced cell migration and invasion….” in the “Abstract and Results” parts according to the reviewer’s suggestion. 

6) In "Introduction" the statement "...leads to depolymerization and division of the Golgi..." is incorrect; the Golgi is not a polymer and it does not divide. 

Response: Thank the reviewer for pointing out the issue. The statement provided is misleading. Therefore, we refer to the original literature to modify the description as “GRASP65 is regulated by Cdc2 and PLK-1 during cell mitosis, which leads to GRASP65 deoligomerization and then Golgi unstacking” in the revised manuscript. 

7) The sequence of each GRASP65 siRNA oligonucleotide should be provided, as well as a description of each of the plasmids used for the overexpression of GRASP65.

Response: Thank the reviewer for the useful suggestion. We have provided the sequence of every GRASP65 siRNA oligonucleotide and the plasmid for overexpression of GRASP65 from NCBI (GRASP65/GORASP1(human): NM_031899.3), which were all designed and synthesized by Shanghai GenePharma Co., Ltd. 

8) The results shown in Fig 1A should be removed, because similar results are already published.

Response: Thank the reviewer for the useful suggestion. We have deleted the results in Fig 1A according to the reviewer’s suggestion. 

9) In "Results" the statement "...and almost completely blocked the closure..." is incorrect; at the most, the treatments reduced SKOV3 migration or invasion to ∼25% the respective levels observed in control conditions.

Response: Thank the reviewer for pointing out the issue. We have modified the results as “As shown in Fig. 1A and 1B, 40 μM DHM suppressed approximately 50% of wound closure in A2780 cells, and significantly reduced the closure of the wound after DHM treatment at both 80 μM and 120 μM in SKOV3 cells” in the revised manuscript. 

10) The title of Fig. 1 legend is incorrect, because DHM did not inhibit cell viability, cell migration and cell invasion; it reduced the extent of these processes.

Response: Thank the reviewer for pointing out the issue. We have modified the title of Fig.1 as “DHM reduced cell migration and invasion in SKOV3 and A2780 cells” in the revised manuscript. 

11) In "Results" the statement "...DHM downregulated the expression of GRASP65 in a concentration-dependent manner, followed by activation of Caspase-3"... is odd. The published data indicate that the process is the opposite, meaning that during apoptosis the activation of caspase-3 results in cleavage of GRASP65 and thus in GRASP65 downregulation.

Response: Thank the reviewer for pointing out the incorrect description. According to the reviewer’s suggestion, we have modified the statement as “In comparison to the control group, DHM induced cell apoptosis as shown in Fig.2, followed by the downregulation of GRASP65 expression in a concentration-dependent manner in Fig. 3C/D.” in the manuscript. 

12) In the legend of Fig. 3 " #p < 0.05, ##p < 0.01 vs the DHM group" should be removed.

Response: Thank the reviewer for pointing out the issue. We have deleted the unnecessary part in the legend of Fig.3. 

13) In "Results" the title "GRASP65 was essential for the anti-cancer effects of DHM in A2780 cells" is incorrect, because the data does not show at all that GRASP65 is essential for the effects of DHM.

Response: Thank the reviewer giving us the useful suggestion. Frankly speaking, it’s not so clear to clarify the relation between the GRASP65 suppression and DHM-induced apoptosis at first. According to the suggestion, we have modified the title as “DHM-induced caspase-3 activation was crucial for suppression of GRASP65 expression in SKOV3 and A2780 cells” in the revised manuscript. In addition, we have read and cited more reference to testify our findings. 

14) In the same section of "Results" the statement "... cells were pre-treated with a specific caspase-3 inhibitor, Ac-DEVD-CHO, for 30 min to suppress the effects of DHM..." is odd, because the experiment should not have been designed to suppress the effects of DHM, but instead to evaluate the contribution of caspase-3 in the effects of DHM.

Response: Thank the reviewer for pointing out the misleading issue. We have modified the statement as “OCs were pre-treated with a specific caspase-3 inhibitor, Ac-DEVD-CHO, for 30 min to suppress the activity of caspase-3 to evaluate the contribution of caspase-3 in the effects of DHM” in the revised manuscript. 

15) The title of Fig. 4 legend is odd; please revise.

Response: Thank the reviewer giving us the useful suggestion. We have revised the title of Fig.4 as “Caspase-3 activation is crucial for suppression of GRASP65 expression in SKOV3 and A2780 cells” in the revised manuscript. 

16) In "Results" the statement "These results suggested that downregulation of GRASP65 could promote DHM-induced inhibition of cell viability and cell migration" is at least speculative, and should be revised. The data indicate that the effects of GRASP65 RNAi and DHM treatment are additive and thus very unlikely to be mechanistically related.

Response: We really appreciate the reviewer for providing these suggestions due to the inaccurate statement. According to the reviewer’s suggestion, we have modified the statement as “These results suggested that GRASP65 depletion using siRNA combined with DHM treatment had an additive effect during DHM-induced inhibition of cell viability and cell migration” in the “results” part and restated again in the “Discussion” part. 

17) Please revise the magnitude informed of the scale bar in the legend of Fig. 6 as it seems very similar to that of Figures 2 and 3.

Response: Thank the reviewer for pointing out the issue. We have revised the results in Fig. 6 and provided the statistical analysis of cell viability and migration. 

18) Please, provide a rationale for not performing all the subsequent analyses in SKOV3 cells.

Response: Thank the reviewer giving us the useful suggestion. In fact, we performed the experiment design including the analysis in two cell lines at first. Until now, we have achieved some results all provided in the manuscript and we thought the findings could indicate the present issue. Surely, we would like to support the subsequent results in the future. 

19) Please, explain why it was not performed the analysis of the levels of p-p38 in cells transfected with siGRASP65. 

Response: Thank the reviewer giving us the useful suggestion. Firstly, our results showed that there was no obvious difference in the phosphorylation of p38 levels among the DHM treated groups as shown in Fig.7A in the revised manuscript. However, we tested again the levels of p-38 in cells transfected with siGRASP65 according to the reviewer’s suggestion, and the results showed no difference among the four groups as shown in Fig.7B. 

20) In "Discussion" the statement "The Golgi is essential for the endoplasmic reticulum and mitochondria..." is odd; please revise.

Response: Thank the reviewer giving us the useful suggestion. We have deleted the sentence due to the misunderstanding. 

21) In "Discussion" the statement "GRASP65, a peripheral Golgi membrane protein, is required for mitotic or apoptotic Golgi fragmentation when specifically cleaved by caspases" is odd; please revise.

Response: Thank the reviewer for pointing out the issue. We have modified the statement as “Cleavage of GRASP65 by caspase-3 correlates with Golgi fragmentation [7], and the fragmentation partially prevented by the expression of a caspase-resistant form of GRASP65 during apoptosis [29, 39]” and added relative reference as shown in the “Discussion” section. 

22) In "Discussion" the statement "...Golgi execution phase of apoptosis..." is odd; please revise.

Response: Thank the reviewer for pointing out the issue. We have modified the section as “ Additionally, the Golgi-localized caspase-2 and -3 are generally accepted as central players in the execution phase of apoptosis, as they mediate cleavage of several golgins and GRASPs, including GM130[41] and GRASP65[29]”in the revised manuscript and added the reference. 

23) In "Discussion" the statement "This implies that the Golgi is a potential therapeutic target, as Golgi disruptive agents may facilitate Golgi fragmentation and induce apoptosis" is a hypothesis already tested by several groups with several published examples in the literature. The authors should discuss their findings in the context of the published data.

Response: Thank the reviewer giving us the useful suggestion. We have modified the statement and added more references to further discuss our findings as shown in the revised manuscript.

24) In "Discussion" the statement "...Golgi formation may be carcinogenic, or a consequence of cancer progression" is immensely odd; please provide more explanations or revise.

Response: Thank the reviewer for pointing out the issue. We have deleted the misleading statement. 

25) In "Discussion" the statement "... inhibiting cleaved caspase-3 can block apoptotic cell death and increasing Caspase-3-like protease activity may be responsible for the delayed cell death" is odd; please revise.

Response: Thank the reviewer for pointing out the issue. We have modified as “Furthermore, we found inhibition of caspase-3 activity by Ac-DEVD-CHO could mitigate DHM-induced cell apoptosis to delay or reduce cell death, followed by an increase of GRASP65 level” in the revised manuscript. 

26) In "Discussion" the statement "... DHM may activate caspase-3, which then cleaves and reduces GRASP65 expression to promote cell apoptosis" is an overstatement, because the data do not support the conclusion that the proapoptotic effects of DHM are mediated by the reduction in the levels of GRASP65.

Response: We really appreciate the reviewer for providing these suggestions due to the inaccurate statement. We have modified the statement as “Therefore, we speculated that DHM might activate caspase-3, which is closely related to the suppression of GRASP65 expression during DHM-induced apoptosis. However, the significance of GRASP65 suppression and its relationship with DHM-mediated effects remain obscure” in the revised manuscript. 

27) In "Discussion" the statement "... activated caspase-3-mediated cleavage and the reduction of GRASP65 was crucial for DHM-induced cell apoptosis" is redundant, and, again, is not supported by the data.

Response: Thank the reviewer giving us the useful suggestion. We have deleted the redundant sentence. 

28) The complete "Conclusion" section should be revised, because many statements are misleading (considering that some conclusions are incorrect).

Response: Thank the reviewer giving us the useful suggestion. We have removed the “Conclusion” part and further discuss our findings according to the published data. 

29) In Fig. 3A and 3B the expected effect of DHM is the fragmentation of the Golgi apparatus. Because DHM resulted in a decrease in the levels of GRASP65, a different Golgi resident protein should be analyzed by immunofluorescence. Also, to diagnose Golgi apparatus fragmentation (instead of Golgi vesiculation) simultaneous immunofluorescence of at least cis and trans Golgi resident proteins should be provided. Higher magnification of the Golgi ribbon in control-treated cells and DHM-treated cells should be also included to properly assess Golgi fragmentation.

Response: Thank the reviewer giving us the useful suggestion. In the present study, we want to indicate that DHM could induce Golgi apparatus fragmentation as well as a decrease in GRASP65 level, and then result in cell apoptosis in OCs. At present, we didn’t perform IF analysis of another Golgi resident protein for other reasons. We have designed experiment to confirm GF and the relationship between GF and Golgi function in DHM-mediated effects and we are doing experiment using confocal microscopy and TEM, no better results yet. However, we need some time to solve the issue. We would like to provide the subsequent results at any time. 

Finally, we would like to appreciate the editor and the reviewers again for positive comments and constructive suggestions that are benefit for the improvement of our manuscript. We believe our revised manuscript is greatly improved and it will be satisfactory with you and reviewers. Meanwhile, we hope our revised manuscript can meet the requirements of your journal for the publication in Plos one. 

Please do not hesitate to contact me if you still have any questions and I am looking forward to hearing from you. 

Sincerely yours,

Fengjie Wang

---

## [Decision Letter · Decision Letter 1]

6 Nov 2019

Golgi reassembly and stacking protein 65 downregulation is required for the anti-cancer effect of dihydromyricetin on human ovarian cancer cells

PONE-D-19-21799R1

Dear Dr. luo,

We are pleased to inform you that your manuscript has been judged scientifically suitable for publication and will be formally accepted for publication once it complies with all outstanding technical requirements.

With kind regards,

Yi-Hsien Hsieh, Ph.D.

Academic Editor

PLOS ONE

Additional Editor Comments (optional):

Reviewers' comments:

Reviewer's Responses to Questions

**Comments to the Author**

1. If the authors have adequately addressed your comments raised in a previous round of review and you feel that this manuscript is now acceptable for publication, you may indicate that here to bypass the “Comments to the Author” section, enter your conflict of interest statement in the “Confidential to Editor” section, and submit your "Accept" recommendation.

Reviewer #1: All comments have been addressed

2. Is the manuscript technically sound, and do the data support the conclusions?

Reviewer #1: Yes

3. Has the statistical analysis been performed appropriately and rigorously? 

Reviewer #1: Yes

4. Have the authors made all data underlying the findings in their manuscript fully available?

Reviewer #1: Yes

5. Is the manuscript presented in an intelligible fashion and written in standard English?

Reviewer #1: Yes

6. Review Comments to the Author

Reviewer #1: The authors properly answered all my raised questions and the revised version in now suitable for publication.

7. PLOS authors have the option to publish the peer review history of their article (what does this mean?). If published, this will include your full peer review and any attached files.

Reviewer #1: No

---

## [Editor Report · Acceptance letter]

13 Nov 2019

PONE-D-19-21799R1 

Golgi reassembly and stacking protein 65 downregulation is required for the anti-cancer effect of dihydromyricetin on human ovarian cancer cells 

Dear Dr. Luo:

I am pleased to inform you that your manuscript has been deemed suitable for publication in PLOS ONE. Congratulations! Your manuscript is now with our production department. 

With kind regards,

on behalf of

Dr Yi-Hsien Hsieh 

Academic Editor

PLOS ONE